# Exploring *Echinops polyceras* Boiss. from Jordan: Essential Oil Composition, COX, Protein Denaturation Inhibitory Power and Antimicrobial Activity of the Alcoholic Extract

**DOI:** 10.3390/molecules28104238

**Published:** 2023-05-22

**Authors:** Hazem S. Hasan, Ashok K. Shakya, Hala I. Al-Jaber, Hana E. Abu-Sal, Lina M. Barhoumi

**Affiliations:** 1Department of Plant Production and Protection, Faculty of Agricultural Technology, Al-Balqa Applied University, Al-Salt 19117, Jordan; hazem@bau.edu.jo; 2Pharmacological and Diagnostic Research Center, Faculty of Pharmacy, Al-Ahliyya Amman University, Amman 19328, Jordan; 3Chemistry Department, Faculty of Science, Al-Balqa Applied University, Al-Salt 19117, Jordanlinabrhoumi@bau.edu.jo (L.M.B.)

**Keywords:** *Echinops polyceras*, essential oil, GC/MS, aliphatic hydrocarbons and their derivatives, COX-1, COX-2, antimicrobial activity, protein denaturation

## Abstract

In this article, we present the first detailed analysis of the hydro-distilled essential oil (HDEO) of the inflorescence heads of *Echinops polyceras* Boiss. (Asteraceae) from the flora of Jordan, offering observations at different growth (pre-flowering, full-flowering and post-flowering) stages. Additionally, we investigated the methanolic extract obtained from the aerial parts of the plant material at the full flowering stage in order to determine its inhibitory activity in terms of COX and protein denaturation and evaluate its antimicrobial effects against *S. aureus* (Gram-positive) and *E. coli* (Gram-negative) bacteria. Performing GC/MS analysis of HDEO, obtained from the fresh inflorescence heads at the different growth stages, resulted in the identification of 192 constituents. The main class of compounds detected in these three stages comprised aliphatic hydrocarbons and their derivatives, which amounted to 50.04% (pre-flower), 40.28% (full-flower) and 41.34% (post-flower) of the total composition. The oils also contained appreciable amounts of oxygenated terpenoids, primarily sesquiterpenoids and diterpenoids. The pre-flowering stage was dominated by (2E)-hexenal (8.03%) in addition to the oxygenated diterpene (6*E*,10*E*)-pseudo phytol (7.54%). The full-flowering stage primarily contained (6*E*,10*E*)-pseudo phytol (7.84%), β-bisabolene (7.53%, SH) and the diterpene hydrocarbon dolabradiene (5.50%). The major constituents detected in the HDEO obtained at the post-flowering stage included the oxygenated sesquiterpenoid intermedeol (5.53%), the sesquiterpene hydrocarbon (*E*)-caryophyllene (5.01%) and (6*E*,10*E*)-pseudo phytol (4.47%). The methanolic extract obtained from air-dried aerial parts of *E. polyceras* displayed more COX-2 inhibition than COX-1 inhibition at a concentration level of 200 µg/mL. The extract exhibited a capacity to inhibit protein denaturation that was comparable with respect to the activity of diclofenac sodium and displayed moderate levels of antimicrobial activity against both bacterial species. The current results demonstrate the need to perform further detailed phytochemical investigations to isolate and characterize active constituents.

## 1. Introduction

The Asteraceae family, commonly known as Compositae, is among the largest families in the plant kingdom, with almost 1620 genera and 23,600 species distributed in almost all habitats worldwide, except underwater. Plants belonging to this family are commonly described as herbs, shrubs and trees. *Echinops*, one of the genera belonging to this family, is known to comprise approximately 120–130 distinct species [1,2,3]. *Echinops* plants, described as perennials, annuals and biennials, are known to grow wild in Eastern and Southern Europe, including the Mediterranean region, as well as in North Africa, the Afrotropical realm and the continent of Asia [1,4].

Several species of the *Echinops* genus have been long used in traditional medicine for the treatment of many ailments, primarily those illnesses related to inflammation, pain and fever [5,6]. Members of this genus have been extensively utilized in Ethiopia to treat of various ailments such as migraine, heart pain, diarrhea, hemorrhoid and intestinal worms [7]. In Ethiopian herbal medicine, chewing the roots of *Echinops kebericho* is prescribed to alleviate stomachache in humans, while roots decoction is employed to cure intestinal diseases in cattle [8]. Additionally, the flower heads and roots of many *Echinops* species find use in Arabian, Cameroonian, Chinese and Indian folk medicine to treat renal disorders and kidney stones [9], reduce asthma attacks [10], stimulate milk secretion [11], and alleviate sexual disability [12]. This wide range of bioactivities has been basically attributed to the wide spectrum of secondary metabolites, including the common members of this genus, i.e., terpenoids, sterols, flavonoids, alkaloids and thiophenes [1,13].

There are three *Echinops* species reported to grow wild in the flora of Jordan. These include *Echinops polyceras* Boiss., *Echinops spinosissimus* Turra. and *Echinops glaberrimus* DC. [14]. *E. polyceras* Boiss (Synonym: *E. blancheanus* Boiss., *E. spinosus* auct. non L.) is the most common species reported to grow wild in the flora of Jordan. This plant is commonly known as globe thistle. This species can best be described as a perennial, spiny, hairy plant that is 60–100 cm long. The leaves are long, dissected with spiny segments; their flower heads are spherical, spiky, 4–5 cm in diameter and characterized by their pale blue color. In the Arab region, primarily in Jordan, this plant is known as “chouk el Jemel” and is known to grow wild in waste places and hills of Irbid, Jarash, Al-Salt, Amman, Madaba, Al-Karak, and Al-Tafila. Flowering occurs during the period extending from July to October [13]. The plant is also reported to grow wild in the Mediterranean neighboring countries including Iraq, Lebanon, Syria, Jordan, Palestine and Saudi Arabia, as well as those located along the coast of the North African coast of the Mediterranean Sea. 

*E. polyceras* has been applied in the folk medicine of many cultures in order to treat a wide variety of ailments. In the Mediterranean region, a decoction prepared from the roots is used to treat renal diseases and kidney stones [9]. In Saudi Arabia, the plant is used for the treatment of gastric pain, indigestion and spasmolytic problems [15]. In Algeria, the plant has been described as finding use in the treatment of dysmenorrhea and prostatism [16]. Previous phytochemical screening studies performed on this species revealed its richness in different classes of secondary metabolites, including flavonoids, sterols, terpenoids and quinolone alkaloids [1,17]. Despite the importance of this species, a thorough literature survey revealed that the plant has not previously been investigated in Jordan, neither for its phytochemical constituents (volatile and nonvolatile secondary metabolites) nor its bioactivity potentials. Accordingly, the current study aims to identify the chemical profile of the hydro-distilled essential oil (HDEO), obtained from the inflorescence heads of Jordanian *E. polyceras*, at different growth stages. In addition, we report on the determination of COX-1 and COX-2, in addition to the protein denaturation inhibitory activities and the antimicrobial potential of the methanolic extract (EPM) obtained from the aerial portions of *E. polyceras*.

## 2. Results

### 2.1. GC/MS Analysis

The results for the GC/MS analysis of the essential oils obtained from *E. polyceras* inflorescence heads at the different growth stages are shown in Table 1. Figure 1 indicates the different classes of volatile principles detected at the different growth stages in the analyzed essential oils. GC/MS chromatograms are available in Appendix A.

### 2.2. Bioactivity Results

#### 2.2.1. COX-1, COX-2 and Protein Denaturation Inhibitory Activity

The methanolic extract obtained from the air-dried aerial parts of *E. polyceras* (EPM) was investigated for its capacity to inhibitory effects against COX-1, COX-2 and protein denaturation. Results are shown in Table 2. 

#### 2.2.2. Antimicrobial Activity

The EPM extract was assayed for its antimicrobial activity against *S. aureus* and *E. coli*. Results are shown in Table 3 and Figure 2.

## 3. Discussion

GC/MS analysis of the HDEO obtained from the inflorescence parts of *E. polyceras* collected at the different growth stages resulted in the identification of a total of 192 compounds (Table 1). Figure 2 reveals the different classes of compounds detected at the different growth stages. The results of this analysis revealed simple nonterpenic aliphatic hydrocarbons and their derivatives (AH&D) to be the main class that dominated the composition in all growth stages.

During the pre-F stage, HDEO contained good amounts of oxygenated diterpenes (OD, 14.05%) in addition to AH&D (50.04%). The primary individual constituents detected in this stage included (2E)-hexenal (8.03%), (6*E*,10*E*)-pseudo phytol (7.54%), hexadecanoic acid (4.61%), *n*-hexanol (4.06%), *n*-nonanal (3.47%), 3-α-14,15-dihydro-manool oxide (3.43%) and heptacosane (2.53%). Additionally, aromatic compounds were detected in appreciable amounts (9.48%) and were represented mainly by 2-pentyl furan (6.32%). 

During the full-flowering and post-flowering stages (full-F, post-F, respectively), AH&D had the highest contribution to both essential oils composition but was detected at slightly lower concentration levels as compared to what occurred at the pre-F stage (40.28%, 41.38%, respectively). The major chemical constituents detected in the EO at the full-F stage were (6*E*,10*E*)-pseudo phytol (7.84%), β-bisabolene (7.53%), dolabradiene (5.50%), intermedeol (4.14%) and tricosanal (3.78%). The principal compounds detected during the post-F stage were intermedeol (5.53%), (E)-caryophyllene (5.01%), (6*E*,10*E*)-pseudo phytol (4.47%), caryophyllene oxide, (3.27%) linalool (3.13%) and hexadecanoic acid (2.04%). 

Via a thorough investigation of the literature, it was revealed that the essential oil composition of the flowering heads of *E. polyceras* had never previously been investigated before. Previous studies concentrated on evaluating the constituents of the essential oils extracted from the roots of most *Echinops* species, including *E. polyceras* [18]. Recently, the study of Belabbes et al. [18] identified 5-(but-1-yn-3-enyl)-2,2’bithiophene and α-terthienyle (54.4 and 26.3%, respectively) as the primary constituents of the essential oil extracted from the roots of *E. spinousus* from Algeria. These two compounds were also detected in the essential oil obtained from the roots of *E. spinousus* of Tunisian origin (21.334%, 18.024%., respectively) [19]. These studies confirm that the essential oils extracted from different plant organs can have quite different compositions. Other factors can also affect the essential oil composition, including harvesting time, extraction method, soil type and many other environmental factors.

The chemical compositions of the essential oils of other *Echinops* species were investigated [20,21,22,23,24,25]. In these studies, the essential oils were extracted from a variety of different organs including the roots and aerial organs (stems, leaves, flowers, tubers). Table 4 summarizes the results of these studies and compares them to our current investigation [20,21,22,23,24,25]. These data shown in Table 4 reveal great qualitative and quantitative differences between the constituents of the EOs among the different *Echinops* species. It was noticed that some chemical constituents were common to all species, including 1,8-cineole, E-caryophyllene, caryophellene oxide and hexadecanoic acid.

In the current study, the methanolic extract (EPM) obtained from the air-dried aerial parts of *E. polyceras* was investigated for its COX-1, COX-2 and protein denaturation inhibitory effects in addition to its antimicrobial potential against the Gram-positive and Gram-negative bacteria. The preliminary screening results indicated that the extract displayed more COX-2 inhibition than COX-1 inhibition. The percentage inhibition of COX-2 was 96.4% and 99.0% at 200 µg/mL and 400 µg/mL EPM concentration levels, respectively. The standard drug used, celecoxib, produced an 88.6% inhibition under the same experimental conditions. It has been widely reported that the COX-1 enzyme is responsible for the synthesis of the constitutional prostaglandins that are responsible for maintaining the integrity of the stomach lining and kidney functions. The inhibition of COX-1 produced certain side effects such as bleeding and ulceration. In this study, it was observed that EPM extracted at a concentration level of 200 μg/mL inhibited only 65.0% of COX-1 enzyme, while the standard reference compound SC-560 (5 ng/mL) inhibited 50.2% COX-1. These findings suggest that fractionation of the extract is required to explore the selective COX-2 activity of the isolated compounds. 

Moreover, the EPM extract displayed moderate antibacterial activity against Gram-positive *S. aureus.* The extract showed weak antibacterial activity against the Gram-negative *E. coli* (Figure 1), with lower levels of activity than those observed for the extract obtained from the *E. polyceras* from Saudi Arabia [26]. 

## 4. Materials and Methods

### 4.1. Plant Material

The plant material was collected from the area surrounding Al-Balqa Applied University, (32′02′11″76″ N; 35′43′43″82″ E), Al-Salt governorate, Jordan, during the summer season of the year 2021. The inflorescence heads of the plant material were collected at the pre-flowering (*pre*-F), full-flowering (full-F) and post-flowering (post-F) stages. The taxonomic identity of the plant was confirmed by Prof. Dr. Hala I. Al-Jaber, Department of Chemistry, Faculty of Science, Al-Balqa Applied University, Al-Salt, Jordan. A voucher specimen (No: Ast/Ep/2021) was deposited at the herbarium of the Faculty of Science (Natural Products Laboratory Herbarium), Al-Balqa Applied University, Al-Salt, Jordan.

### 4.2. Hydro-Distillation and Extraction of Essential Oils

Essential oils (EOs) were extracted from fresh inflorescence heads of *E. polyceras* at different growth stages and according to the procedure described in the literature [27,28]. Briefly, a 300 g sample of fresh inflorescence heads collected at each flowering stage was coarsely powdered and then subjected to hydro-distillation for 3 h in a Clevenger-type apparatus. The obtained essential oil (HDEO) from each growth stage was extracted (twice) with GC-grade *n*-hexane, dried using anhydrous MgSO_4_, and then stored in amber glass vials at 4 °C until analysis was performed. 

### 4.3. GC-MS Analysis

GC/MS analysis was done according to the procedure previously described in the literature [29,30]. The analysis was performed on a Shimadzu QP2020 GC-MS equipped with GC-2010 Plus (Shimadzu Corporation, Kyoto, Japan) with a split–splitless injector, utilizing a DB5-MS fused silica column (5% phenyl, 95% polydimethylsiloxane, 30 m × 0.25 mm, 25 µm film thickness). Briefly, a linear temperature program was used to separate the different components. The oven temperature was set to 50 °C for 1 min, after which temperature programming was applied at 7 °C/min. The heating rate started from 50 °C (initial temperature) to 280 °C (final temperature); this was then held at 280 °C for 10 min, and the total run time was 44 min. The injector temperature was 260 °C with a split ratio of 20:1; an injection volume of 1 µL; a carrier gas: helium (flow rate 1.50 mL/min); and a flow control mode: pressure, 88.3 kPa. MS source temperature/detector temperature: 240 °C; interface temperature: 250 °C; ionization energy (EI): 70 eV; scan range 35–500 amu; scan speed 1666. The solvent cut was 3 min, while these data were acquired in 4.5 min. These data were collected using Windows-based Lab-Solution GC-MS version 4.45SP1 Software. The mass spectra of isolated components were compared to those reported in ADAMS-2007 and NIST 2017 mass spectrometry libraries. To confirm the identified compound, a comparison was performed between reported values and relative retention indices (RRI) with reference to *n*-alkanes (C_8_–C_30_) in addition to these data published in the literature [31]. 

### 4.4. Preparation of the Alcoholic Extract

The alcoholic extract was prepared according to the procedure described in the literature with slight modifications [27,32]. Air-dried aerial parts of *E. polyceras* (20 g) were ground down to fine powder and then soaked in methanol (400 mL) at room temperature (3 × 24 h). The solvent was then evaporated under reduced pressure at 55 °C. The obtained methanol extract (EPM, 2.2479 g; yield: 11.24%) was then used for bioactivity screening. 

### 4.5. COX-1 and COX-2 Inhibitory Activity

To evaluate the COX activity and to judge the NSAID activity of the methanolic extract (EPM), a COX-1 inhibitory screening assay kit containing SC-560 as standard (Fluorometric, ab204698, Abcam, Tokyo, Japan) and COX-2 inhibitor screening kit (Fluorometric, ab283401, Abcam, Tokyo, Japan) containing celecoxib as standard were used as per the instruction provided with the kit without any modification [33,34,35]. Two different concentrations of the EPM extract were prepared for initial screening in the supplied buffer. The buffer solution was used as a control. 

### 4.6. Inhibition of Protein Denaturation

The procedure described in [35,36], was used to estimate the inhibition of protein denaturation with slight modifications. Different solutions were prepared for the assay, including the test solution (EPM extract), test control, product control and standard solution. All solutions were prepared using a buffer with pH 6.3. These prepared samples were kept at 37 °C for 20 min; then, they were incubated at 50 °C for another 20 min. The absorbance was measured at 416 nm using a synergy HTC multimode reader (Viotek, Winooski, VT, USA). Absorbance measurements were performed after all samples had cooled down. The percent inhibition of protein denaturation was calculated using the given formula:$$\mathrm{Percent}\mathrm{Inhibition}=100-(\frac{\;Abs\;of\;test\;solution-Abs\;of\;Product\;control}{Abs\;of\;Test\;control}\times 100)$$

### 4.7. Antimicrobial Activity

Gram-positive (*S. aureus* ATCC 6538) and Gram-negative (*E. coli* ATCC 8739) bacteria were used to assess the antibacterial activity of the EPM extract and moxifloxacin (standard drug) according to the procedure described in the literature [34]. Nutrient agar was added to a sterile Petri disc (9 cm, diameter). The plates were inoculated with bacterial culture using sterilized cotton swaps. Subsequently, 6 mm wells were created with sterilized 6 mm surgical punches. The wells were filled with 60 µL of extract solution (5000, 2500 or 1250 µg/mL) and moxifloxacin standard solution (40 µg/mL). A 5% DMSO solution was used as a control. The plates were incubated at 37 °C for 24 h. Duplicate runs of each experiment were produced.

### 4.8. Statistical Analysis

The results obtained in the present study are expressed as mean ± standard deviation (SD). For the statistical analysis of the experimental data, Graph-Pad Prism 5 (Graph-Pad Software, San Diego, CA, USA) was used.

## 5. Conclusions

This is the first study to report on the chemical composition of hydro-distilled essential oil obtained from the inflorescence heads of Jordanian *E. polyceras* at different growth stages. When comparing our current results with those reported in the literature, it can be concluded that the essential oil obtained from flowering heads of *E. polyceras* had a quite different composition than that obtained from the roots. The EO was rich in chemical constituents such as (6*E*,10*E*)-Pseudo phytol, dolabradiene, isoamyl dodecanoate, intermedeol, β-bisabolene, linalool, (*E*)-caryophyllene, caryophyllene oxide, nerol, β-elemene, hexaxecanoic acid and iso-longifolol. The methanolic extract obtained from the aerial parts of *E. polyceras* showed significant COX-1, COX-2 and protein denaturation inhibitory activities, but moderate antimicrobial activity, against *S. aureus* and *E. coli*. The current results of this study encourage researchers to undertake further detailed phytochemical investigations of the alcoholic extract in order to isolate and characterize its active constituents.

## Figures and Tables

**Figure 1 molecules-28-04238-f001:**
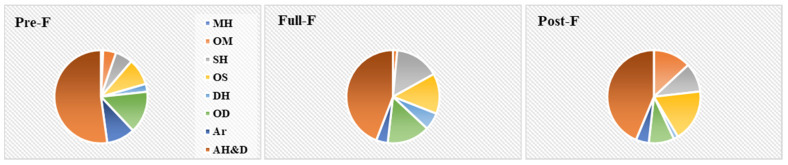
Different classes of compounds detected in the HDEOs of *E. polyceras* from Jordan collected at different growth stages *viz.* Pre-F: pre-flowering, Full-F: full-flowering and Post-F: post-flowering.

**Figure 2 molecules-28-04238-f002:**
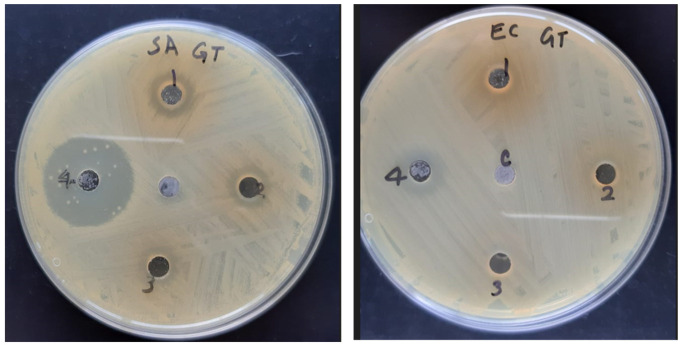
Antibacterial activity of EPM extract: well 1:5000 µg/mL; well 2:2500 µg/mL; well 3:1250 µg/mL, and moxifloxacin (well 4:40 µg/mL) against *S. aureus* (**left**) and *E. coli.* (**right**). Control (5% DMSO) is marked “C”.

**Table 1 molecules-28-04238-t001:** GC/MS analysis of the HDEO obtained from inflorescence heads of fresh *E. polyceras* at different growth stages.

No	RI ^exp^	RI ^Theor^	Compound	Class	% Composition		
					PF	FF	Post-F
1	846	841	3-methyl-Pentanol	AH&D	0.09	3.00	-
2	850	850	(2*E*)-Hexenal	AH&D	8.03	-	-
3	851	853	(3*E*)-Hexenol	AH&D	-	-	0.23
4	861	862	(2*E*)-Hexenol	AH&D	0.96	-	0.15
5	865	870	*n*-Hexanol	AH&D	4.06	-	0.86
6	865	867	(2Z)-Hexenol	AH&D	-	0.08	-
7	884	888	Ethyl pent-4-enoate	AH&D	-	0.27	-
8	886	889	4-Heptanol	AH&D	-	0.15	-
9	888	892	2-Heptanone	AH&D	0.13	-	-
10	898	895	(4*Z*)-Heptenal	AH&D	-	0.13	-
11	899	900	*n*-Nonane	AH&D	0.14	-	-
12	904	902	Heptanal	AH&D	0.43	0.10	0.03
13	916	916	(2*E*,4*E*)-Hexadienol	AH&D	0.12	2.66	0.01
14	921	923	2-methyl-4-Heptanone	AH&D	-	0.59	-
15	932	944	5-methyl-3-Heptanone	AH&D	-	0.34	-
16	948	943	3-methyl-Cyclohexanol	AH&D	-	2.30	-
17	954	952	3-methyl-Cyclohexanone	AH&D	-	0.24	-
18	957	954	(2*E*)-Heptenal	AH&D	0.12	-	0.03
19	963	960	Benzaldehyde	AH&D	-	-	0.03
20	968	966	*n*-Heptanol	AH&D	-	-	0.03
21	973	973	Hexanoic acid	AH&D	-	-	0.05
22	978	979	1-Octen-3-ol	AH&D	-	-	0.10
23	984	986	(3*E*)-Octen-2-ol	AH&D	-	-	0.05
24	990	988	2-Pentyl furan	Ar	6.32	-	0.15
25	999	1002	δ-2-Carene	MH	0.75	-	0.04
26	1004	998	*n*-Octanal	AH&D	0.14	-	0.10
27	1012	1016	(2*E*,4*E*)-Heptadienol	AH&D	-	-	0.09
28	1027	1021	3-methyl-1,2-Cyclopentanedione	AH&D	-	-	0.03
29	1034	1031	Eucalyptol	OM	-	-	0.08
30	1038	1035	(3*E*)-Octen-2-one	AH&D	-	-	0.09
31	1045	1042	Benzene acetaldehyde	Ar	-	-	0.05
32	1055	1054	Prenyl isobutyrate	AH&D	0.63	-	0.11
33	1059	1054	(2*E*)-Octen-1-al	AH&D	0.27	-	0.26
34	1070	1068	*n*-Octanol	AH&D	-	-	0.92
35	1082	1084	(2*Z*)-Hexenal diethyl acetal	AH&D	-	-	0.05
36	1087	1086	*trans*-Linalool oxide	OM	-	-	0.07
37	1090	1090	2-Nonanone	AH&D	-	-	0.04
38	1094	1090	Isobutyl tiglate	AH&D	-	-	0.41
39	1095	1098	2-Nonanol	AH&D	0.11	-	-
40	1100	1096	Linalool	OM	1.44	0.45	3.13
41	1105	1100	*n*-Nonanal	AH&D	3.47	0.27	2.17
42	1111	1108	*cis*-Rose oxide	OM	-	-	0.20
43	1116	1116	(2*E*,4*E*)-Octadienol	AH&D	-	-	0.16
44	1124	1119	*trans*-*p*-Mentha-2,8-dien-1-ol	OM	-	-	0.08
45	1127	1133	1-Terpineol	OM	-	-	0.10
46	1139	1140	Nopinone	OM	-	-	0.11
47	1144	1144	*trans*- *p*-Menth-2-en-1-ol	OM	-	-	0.10
48	1147	1146	*trans*-Verbenol	OM	-	-	0.09
49	1151	1154	Camphor	OM	-	-	0.23
50	1155	1154	(2*E*,6*Z*)-Nonadienal	AH&D	0.79	0.13	1.48
51	1156	1153	(3*E*,6*Z*)-Nonadienol	AH&D	-	-	0.12
52	1161	1157	(2*E*)-Nonen-1-al	AH&D	1.85	0.17	2.63
53	1167	1166	(2*Z*)-Nonenol	AH&D	-	-	0.38
54	1172	1169	*n*-Nonanol	AH&D	0.12	-	0.36
55	1176	1169	Borneol	OM	-	-	0.11
56	1184	1177	Terpinen-4-ol	OM	-	-	0.38
57	1189	1182	*p*-Cymen-8-ol	Ar	-	-	0.49
58	1191	1192	2-Decanone	AH&D	-	-	0.04
59	1199	1196	γ-Terpineol	OM	0.45	0.23	1.09
60	1202	1169	Safranal	OM	-	-	0.06
61	1207	1201	*n*-Decanal	AH&D	0.70	0.70	0.87
62	1211	1208	*trans*-Piperitol	OM	-	-	0.14
63	1215	1205	Verbenone	OM	0.13	-	0.35
64	1218	1229	Nerol	OM	1.03	-	1.51
65	1223	1219	β-Cyclocitral	OM	0.24	-	0.33
66	1225	1225	Citronellol	OM	-	-	0.36
67	1230	1229	(*Z*)-Ocimenone	OM	-	-	0.30
68	1240	1232	exo-Fenchyl acetate	OM	-	-	0.06
69	1250	1252	Geraniol	OM	-	-	0.63
70	1259	1263	*cis*-Carvone oxide	OM	-	-	0.07
71	1264	1263	(2*E*)-Decenal	AH&D	0.33	-	0.81
72	1270	1269	*n*-Decanol	AH&D	-	-	0.30
73	1281	1285	Isobornyl acetate	OM	0.22	-	0.08
74	1284	1285	*iso*-Isopulegyl acetate	OM	-	-	0.32
75	1287	1285	Bornyl acetate	OM	-	-	0.14
76	1291	1290	Thymol	Ar	-	-	0.15
77	1297	1293	(2*E*,4*Z*)-Decadienal	AH&D	0.16	-	0.28
78	1300	1300	*n*-Tridecane	AH&D	0.27	0.29	0.55
79	1309	1306	Undecanal	AH&D	0.32	0.19	0.27
80	1313	1309	*p*-vinyl-Guaiacol	Ar	-	-	0.22
81	1322	1321	(2*E*,4*E*)-Decadienol	AH&D	0.78	0.59	1.67
82	1331	1332	Hexyl tiglate	AH&D	-	-	0.08
83	1337	1338	δ-Elemene	SH	0.11	-	0.16
84	1352	1359	Eugenol	Ar	-	-	0.12
85	1363	1361	γ-Nonalactone	AH&D	-	-	0.17
86	1367	1360	(2*E*)-Undecenal	AH&D	-	0.17	0.33
87	1375	1370	*n*-Undecanol	AH&D	-	-	0.11
88	1382	1376	α-Copaene	SH	0.23	0.31	0.27
89	1390	1390	β-Elemene	SH	0.94	1.07	1.30
90	1399	1399	9-Decenyl acetate	AH&D	-	-	0.22
91	1402	1400	Tetradecane	AH&D	-	0.23	0.10
92	1410	1408	Dodecanal	AH&D	0.22	0.24	0.03
93	1414	1412	dihydro-α-Ionone	OM	0.62	0.55	1.86
94	1429	1419	(*E*)-Caryophyllene	SH	3.10	2.45	5.01
95	1434	1440	*trans*-Nerone	OM	-	-	0.04
96	1437	1433	β-Gurjunene	SH	0.23	0.11	0.07
97	1439	1441	Aromadendrene	SH	-	-	0.05
98	1449	1453	Geranyl acetone	OM	0.18	0.13	0.32
99	1461	1466	(2*E*)-Dodecenal	AH&D	0.54	0.62	0.51
100	1463	1454	α-Humulene	SH	0.12	0.28	0.42
101	1469	1470	*n*-Dodecanol	AH&D	-	-	0.12
102	1479	1479	6-nonyl-5,6-dihydro-2H-Pyran-2-one	AH&D	0.76	0.30	2.71
103	1482	1481	methyl-γ-Ionone	OS	0.83	0.68	1.18
104	1486	1488	(*E*)-β-Ionone	OS	0.10	-	0.44
105	1488	1492	δ-Selinene	SH	-	0.50	-
106	1493	1496	Valencene	SH	0.43	0.33	0.20
107	1497	1490	β-Selinene	SH	0.15	0.27	-
108	1497	1492	δ-Selinene	SH	-	-	0.30
109	1500	1500	*n*-Pentadecane	AH&D	-	0.35	0.18
110	1503	1498	α-Selinene	SH	0.10	0.31	0.40
111	1507	1497	Methyl *p*-tert-butylphenyl acetate	Ar	-	0.12	-
112	1511	1505	β-Bisabolene		-	7.53	0.52
113	1513	1510	Tridecanal	AH&D	0.39	-	-
114	1516	1512	α-Alaskene	SH	0.37	0.17	0.62
115	1522	1515	Cubebol	OS	0.29	-	0.55
116	1528	1523	δ-Cadinene	SH	-	0.76	0.12
117	1543	1546	Selina-3,7(11)-diene	SH	-	-	0.09
118	1561	1566	Dodecanoic acid	AH&D	0.49	0.35	0.89
119	1577	1566	(3*Z*)-Hexenyl benzoate	Ar	1.83	1.11	1.61
120	1584	1580	*n*-Hexyl benzoate	Ar	1.32	0.84	1.33
121	1582	1590	Caryophyllene oxide	OS	1.53	2.22	3.27
122	1600	1600	*n*-Hexadecane	AH&D	-	0.44	0.11
123	1604	1604	Khusimone	OS	-	-	0.11
124	1615	1612	Tetradecanal	AH&D	0.32	0.22	0.30
125	1621	1619	2,(7*Z*)-Bisaboladien-4-ol	OS	-	-	0.42
126	1627	1632	γ-Eudesmol	OS	-	-	0.05
127	1633	1637	Caryophylla-4(12),8(13)-dien-5β-ol	OS	-	-	0.39
128	1647	1650	β-Eudesmol	OS	0.40	0.24	1.50
129	1656	1653	α-Eudesmol	OS	-	-	0.31
130	1657	1659	Selin-11-en-4-α-ol	OS	-	0.17	-
131	1665	1660	neo-Intermedeol	OS	0.10	0.29	0.40
132	1669	1663	7-epi-α-Eudesmol	OS	0.23	0.30	0.40
133	1673	1672	*n*-Tetradecanol	OS	0.36	1.11	0.47
134	1677	1666	Intermedeol	OS	4.19	4.14	5.53
135	1681	1676	Cadalene	Ar	-	1.07	-
136	1688	1692	4-Cuprenen-1-ol	OS	0.15	0.18	0.36
137	1691	1689	Shyobunol	OS	-	-	0.18
138	1702	1700	*n*-Heptadecane	AH&D	0.23	0.56	0.54
139	1717	1703	(2*E*)-Tridecenol acetate	AH&D	1.59	0.86	1.83
140	1721	1718	Methyl eudesmate	Ar	-	0.52	0.12
141	1725	1723	Methyl tetradecanoate	AH&D	-	1.71	0.51
142	1733	1729	*iso*-Longifolol	OS	0.41	1.95	0.90
143	1746	1746	γ-Costol	OS	-	0.18	0.49
144	1760	1774	*n*-Pentadecanol	AH&D	0.94	0.91	0.60
145	1766	1767	12-hydroxy-(Z)-Sesquicineole	OS	-	-	0.08
146	1795	1796	Ethyl tetradecanoate	AH&D	-	-	0.10
147	1802	1800	*n*-Octadecane	AH&D	-	0.47	0.12
148	1806	1808	Eudesm-11-en-4-α, 6-α-diol	OS	-	0.70	0.04
149	1819	1817	(2*E*,6*E*)-Farnesoic acid	OS	0.29	0.61	0.16
150	1831	1829	Isopropyl tetradecanoate	AH&D	0.56	-	0.52
151	1843	1845	Isoamyl dodecanoate	AH&D	2.50	1.86	1.83
152	1901	1900	*n*-Nonadecane	AH&D	0.26	0.24	0.34
153	1921	1930	Ambrettolide	AH&D	0.21	-	0.18
154	1926	1921	Hexadecanoic acid, methyl ester	AH&D	1.26	1.44	1.99
155	1964	1960	Hexadecanoic acid	AH&D	4.61	0.53	2.04
156	1984	1974	Dolabradiene	Ar	2.62	5.50	1.48
157	1996	2000	*n*-Eicosane	AH&D	0.20	0.18	0.19
158	2001	2003	Hexadecyl acetate	AH&D	0.24	0.57	0.18
159	2012	1989	Manoyl oxide	OD	0.40	0.90	0.20
160	2017	2017	Phyllocladene	SH	-	-	0.02
161	2024	2024	Isopropyl hexadecanoate	AH&D	0.31	0.37	0.27
162	2033	2010	13-epi-Manool oxide	OD	-	-	0.06
163	2060	2060	13-epi-Manool	OD	-	0.31	0.05
164	2070	2043?	(6*E*,10*E*)-Pseudo phytol	OD	7.54	7.84	4.47
165	2087	2077	*n*-Octadecanol	AH&D	-	-	0.07
166	2094	2085	Methyl linoleate	AH&D	0.45	0.59	0.64
167	2101	2100	*n*-Heneicosane	AH&D	0.82	0.98	0.84
168	2111	2116	Laurenan-2-one	OD	0.26	0.40	0.31
169	2127	2128	Methyl stearate	AH&D	-	0.35	0.26
170	2134	2133	Linoleic acid	AH&D	1.09	1.37	0.60
171	2139	2142	Oleic acid	AH&D	0.29	0.14	0.13
172	2143	2149	Abienol	OD	0.15	-	0.14
173	2165	2173	Linoleic acid ethyl ester	AH&D	-	-	0.08
174	2180	2189	1-Docosene	AH&D	-	-	0.28
175	2181	2196	Ethyl octadecanoate	AH&D	0.38	-	-
176	2194	2198	Ugandensidial	OD	0.85	0.81	0.43
177	2195	2200	*n*-Docosane	AH&D	0.77	0.67	0.39
178	2209	2209	Octadecanol acetate	AH&D	0.10	0.18	0.05
179	2232	2223	Sclareol	OD	0.12	-	0.11
180	2283	2269	Sandaracopimarinol	OD	0.99	0.70	0.93
181	2300	2297	3-α-hydroxy-Manool	OD	0.32	-	0.08
182	2302	2300	*n*-Tricosane	AH&D	1.38	1.66	1.41
183	2335	2338	3-α-14,15-dihydro-Manool oxide	OD	3.43	2.60	1.53
184	2406	2400	*n*-Tetracosane	AH&D	0.29	0.46	0.31
185	2502	2500	*n*-Pentacosane	AH&D	1.18	1.33	0.78
186	2535	2531	Tricosanal	AH&D	0.79	3.78	0.65
187	2597	2600	Hexacosane	AH&D	0.12	1.08	0.14
188	2702	2700	Heptacosane	AH&D	2.53	1.56	1.60
189	2803	2800	Octacosane	AH&D	0.19	0.41	0.13
190	2815	2790	Squalene	AH&D	-	0.27	-
191	2903	2900	Nonacosane	AH&D	-	0.62	-
192	3101	3101	Untriacontane	AH&D	-	-	0.09
			Monoterpene hydrocarbons (MH)		0.75	-	0.04
			Oxygenated monoterpenes (OM)		4.31	1.36	12.35
			Sesquiterpene hydrocarbons (SH)		5.78	14.08	9.53
			Oxygenated sesquiterpenes (OS)		8.88	12.76	17.22
			Diterpene hydrocarbons (DH)		2.62	5.50	1.48
			Oxygenated diterpenes (OD)		14.05	13.56	8.34
			Aromatics (Ar)		9.48	3.66	4.24
			Aliphatic hydrocarbons & their derivatives (AH&D)		50.04	40.28	41.34
			Total Identified		95.91	91.20	94.50

**Table 2 molecules-28-04238-t002:** COX-1 and COX-2 and protein denaturation inhibitory effects of EPM extract.

	% Inhibition (Mean ± SD)		
	COX-1	COX-2	Protein Denaturation
EPM: (200 µg/mL)	64.98 ± 2.55	96.37 ± 0.85	-
EPM: (400 µg/mL)	-	98.97 ± 0.45	-
SC-560 (5 ng/mL)	50.17 ± 1.25	-	-
Celecoxib *	-	88.63 ± 2.20	-
EPM (250 µg/mL)	-	-	64.34 ± 2.83
Diclofenac sodium (250 µg/mL)	-	-	68.75 ± 1.05

* (5× dilution, as instructed), *n* = 3.

**Table 3 molecules-28-04238-t003:** Antimicrobial activity of EPM extract against *S. aureus* and *E. coli*.

Sample Name	Inhibition Zone (mm)	
	*S. aureus*	*E. coli*
EPM extract (5000 µg/mL)	15	12
EPM extract (2500 µg/mL)	12	10
EPM extract (1250 µg/mL)	10	None
Moxifloxacin (40 µg/mL)	26	13

*n* = two measurements; diameter of well = 6 mm.

**Table 4 molecules-28-04238-t004:** Comparative studies on the chemical composition of the HDEO obtained from the roots, different aerial parts (stems, leaves, flowers, tubers) of several *Echinops* species from different origins around the world and the current study.

Constituents	Current Study			*E. Spinousus* [19]	*E. Ilicifolius* [20]			*E. grijsii* [21]	*E. latifolius* [22]	*E. kebericho* [23]	*E. kebericho* [8]	*E. ritro* [24]	*E. graecus* [25]	*E. ellenbeckii* [25]			
	PF	FF	Pst F	R	L	F	R	R	AP	Tu	R	Infl	Infl	L	S	F	R
(2*E*)-Hexenal	8.03	-	-	-	-	-	-	-	-	-	-	21.4	-	-	-	-	-
𝛽-Pinene	-	-	-	-	-	-	-	3.92	15.56	3.62	-	t	-	-	-	-	-
1,8-Cineole	1.44	0.45	3.13	0.321	-	0.1	-	5.56	19.63	-	tr	16.3	-	-	-	-	-
β-Phellandrene	-	-	-	0.413	-	-	-	-	0.15	10.84	-		-	-	-	-	-
(*Z*)-𝛽-Ocimene	-	-	-	-	0.1	-	-	5.01	18.44	0.83	rt	t	-	-	-	-	-
Linalool	-	-	0.08	0.549	16.4	58.6	0.9	1.54	2.74	-	-	9.8	5.6	-	-	-	-
Geraniol	-	-	0.63	-	8.3	17.4	-	-	-	-	-	t	-	-	-	-	-
p-Cymene	-	-	-	-	-	-	0.9	-	-	0.14	0.1	12.1	t	-	-	-	-
Camphor	-	-	0.23	0.862	0.1	-	0.6	-	1.50	0.16	tr	2.6	6.5	-	-	-	tr
Intermedeol	4.19	4.14	5.53	-	-	-	-	-	-	-	-	-	-	-	-	-	-
(*E*)-Caryophyllene	3.10	2.45	5.01	2.736	-	-	-	3.84	3.58	-	1.7	-	-	-	-	2.30	0.23
β-Bisabolene	-	7.53	0.52	-	-	-	-	0.46	-	-	0.1	-	-	-	-	4.40	-
Caryophyllene oxide	1.53	2.22	3.27	5.217	-	-	-	3.53	0.82	0.40	9.7	-	-	4.93	2.26	-	1.01
γ-Cadinene	-	-	-	27.224	-	2.32	1.52	-	-	-	-	-	-	-	-	-	-
(6*E*,10*E*)-Pseudo phytol	7.54	7.84	4.47	-	-	-	-	-	-	-	-	-	-	-	-	-	-
Dolabradiene	2.62	5.50	1.48	-	-	-	-	-	-	-	-	-	-	-	-	-	-
*n*-Dodecane	-	-	-	-	0.2	10.9	14.5	-	-	-	-	-	t	0.52	0.11	0.90	0.06
Nerol	1.03	-	1.51	-	2.6	5.4	-	-	-	-	-	t	-	-	-	-	-
*n*-Hexadecanoic acid	4.61	0.53	2.04	-	36.2	3.3	7.8	-	-	-	-	2.6	-	-	-	-	7.10
Dehydrocostus lactone	-	-	-	-	-	-	-	-	-	41.83	-	-	-	-	-	-	-
Germacrene B	-	-	-	-	-	-	-	-	-	5.38	-	-	-	-			
Tridecane	0.27	0.29	0.55	-	-	-	0.2	-	-	-	tr	-	-	0.57	0.11	1.50	0.10

AP: aerial parts, L: leaves; S: stem; PF: pre-flower; FF: full flower; F: flower; Infl: Inflorescence; Tu: tubers; Pst-F: post-flower; AP: Aerial Part; R: root.

## Data Availability

Data will be provided upon request.

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
