# Peer review of "Exploring Echinops polyceras Boiss. from Jordan: Essential Oil Composition, COX, Protein Denaturation Inhibitory Power and Antimicrobial Activity of the Alcoholic Extract"

_molecules, 2023, doi:10.3390/molecules28104238_

Round 1

Reviewer 1 Report

Hasan et al. presented a study on the chemical identification of essential oil of the inflorescence heads of Echinops polyceras at different growth stages. Additionally, the extract from flower was investigated for its COX inhibitory activity and antimicrobial activity. In general, this paper is well-written with solid methodlogy and reliable results. I only have several comments and suggestions for the improvement of the paper.

1      It is better to use pie plot instead of bar plot in Figure 1.

2      Through the MS, the number of replicates in each assay is not clear, please provide exact number of the samples.

3      Table 4 is not presented in a standard formation (three line table).

4      Table 2-3, no statistical analysis are performed. Mean± SD should be presented.

5      In 4.3, the details of GC-MS instrument should be added.

6      line 243 should be ‘4.7 Statistical analysis’.

7      The discussion part should not revised to avoid the simple repeating of the results. More references related to the results should be involved.

8      The language should be substantially improved.

 The language should be substantially improved.

Author Response

Please refer Attached file.

Thank you

Reviewer 2 Report

Dear Author.

The MS entitled “Exploring Echinops polyceras Boiss. from Jordan: Exploring Essential Oil Composition, COX, protein denaturation Inhibitory Activity and Antimicrobial Potentials of the Alcoholic Extract” was thoroughly evaluated. My suggestions are:

1. The title contains repetition of the word “exploring”. Change the title.

2. The abstract should be concluded with remarks that reflect the significances of the study.

3. Line 55, 56; space after reference 10 and 12 should be reduced.

4. Introduction. Line 47-58 and 70-76 contains similar literature about biological significance. Combine and remove that. Add research work carried out on the essential oils of Echinops species. Also add some latest citations and literature about essential oils.

https://doi.org/10.3390/ph15101221

https://doi.org/10.1080/10412905.2023.2185310

http://dx.doi.org/10.30848/PJB2019-1(19)

5. Line 55, 56; space after reference 10 and 12 should be reduced.

6.The names of organisms such as microorganism or plants should be italic throughout the MS.

7. Discussion section; para 1, 2 and 3, the author stick to presents their results here rather to compare them with previous studies, in my opinion it is not appropriate to presents results here. It should be properly discussed in scientific ways.

8. The author has discussed the COX-I, COX-II and antibacterial results without coding any relevant reference from the reported studies, it should be re-discussed accordingly.

9. In my opinion the COX-I and COX-II activity using kit only limits the results to outside the living system, the author should add more data using the in-vivo animal model of analgesic or anti-inflammatory study.

10. Table 2; in the COX-I activity, the author use this concentration in ng/mL (SC-560 5ng/mL) while the other concentrations were presented in µg/mL??? the author should mention the reason for it.

11. The constituents recorded in GC-MS of essential oils include higher amounts of phytol in all of them, which is not much bioactive. Define the reason why the essential oils show such higher activities?

12. The central theme of the conclusion of this study is beyond the key results which should be re-modified.

 Extensive editing of English language required before final acceptance of the M.S,

Author Response

Please refer attached file

Round 2

Reviewer 2 Report

The tittle of the manuscript is not appropriate and appealing. So must be changed.The manuscript must be checked from a native speaker.

I recommend the acceptance with subject to correction certificate from a native speaker.

Author Response

Dear Editor

Please find enclosed herewith Revised copy of the manuscript. The title is updated to "Exploring Echinops polyceras Boiss. from Jordan: Essential Oil Composition, COX, protein denaturation Inhibitory Power and Antimicrobial activity of the Alcoholic Extract". I will be grateful if the reviewer can provide suggestion if not accepted by him. The manuscript is corrected by the rapid MDPI English language service. The communication details are available at website.  The graphical abstract is added as separate word file. I hope the Journal will accept the manuscript in the present form. 

Thank you

Sincerely yours

Ashok K. Shakya

Hala I Al-Jaber